# Chemical Conversion of Hardly Ionizable Rhenium Aryl Chlorocomplexes with *p*-Substituted Anilines

**DOI:** 10.3390/molecules26113427

**Published:** 2021-06-05

**Authors:** Martin Štícha, Ivan Jelínek, Mikuláš Vlk

**Affiliations:** 1Department of Chemistry, Faculty of Science, Charles University, 12000 Prague 2, Czech Republic; 2Department of Analytical Chemistry, Faculty of Science, Charles University, 12000 Prague 2, Czech Republic; ijelinek@natur.cuni.cz (I.J.); mikulas.vlk@natur.cuni.cz (M.V.)

**Keywords:** high-resolution mass spectrometry, rhenium complexes, chemical derivatization, coordination chemistry, DFT

## Abstract

Fast and selective analytical methods help to ensure the chemical identity and desired purity of the prepared complexes before their medical application, and play an indispensable role in clinical practice. Mass spectrometry, despite some limitations, is an integral part of these methods. In the context of mass spectrometry, specific problems arise with the low ionization efficiency of particular analytes. Chemical derivatization was used as one of the most effective methods to improve the analyte’s response and separation characteristics. The Schotten–Baumann reaction was successfully adapted for the derivatization of ESI hardly ionizable Re(VII) bis(catechol) oxochlorocomplex. Various alkyl and halogen *p*-substituted anilines as possible derivatization agents were tested. Unlike the starting complex, the reaction products were easily ionizable in electrospray, providing structurally characteristic molecular and fragment anions. DFT computer modeling, which proposed significant conformation changes of prepared complexes within their deprotonation, proved to have a close link to MS spectra. High-resolution MS and MS/MS measurements complemented with collision-induced dissociation experiments for detailed specification of prepared complexes’ fragmentation pathways were used. The specified fragmentation schemes were analogous for all studied derivatives, with an exception for [Re(O)(Cat)_2_PIPA].

## 1. Introduction

In analytical chemistry, derivatization is primarily used to modify an analyte that cannot be analyzed by a particular analytical method or to improve selectivity. Chemical derivatization helps to improve separation characteristics and the sensitivity of detection [1]. Chemical derivatization has played an important role in GC/MS analysis, where derivatization is used to increase volatility, change the analyte’s ionization properties, or affect analyte fragmentation [2]. The most commonly used derivatives in GC/MS are methyl, ethyl, acetyl, or silyl esters of fatty acids. Mostly in situ derivatization methods are used, where the sample preparation and chemical modification take place in one step. Derivatization in liquid chromatography has a different rationale and, therefore, different rules apply when selecting reagents [3]

The goal of chemical derivatization in ESI/MS is to convert the poorly ionizable or non-ionizable substance into an easily detectable one by changing its chemical and physical properties. ESI is considered very sensitive toward polar compounds. However, for low polarity or non-polar compounds it has been considered less satisfactory than APCI. Substances that form ions in a solution are generally well ionizable by ESI. In contrast, APCI is more suitable for ionizing low to medium polar substances containing atoms with high proton affinity [4]. Although ESI/MS is one of the most efficient analytical methods, its use in detecting some low polar rhenium complexes is limited. Problems with the ionization of these molecules are caused by the absence of acidic or basic groups in the structure. The analyte can be chemically modified to increase ionization efficiency by introducing groups capable of protonation or deprotonation [5,6,7,8,9,10,11,12,13,14,15,16,17].

The derivatization potential for ESI-MS in analytical chemistry has been presented in numerous publications [7,8,10,16,17,18,19,20,21,22]. One of the disadvantages of derivatization is the possibility of affecting not only the target analyte but also other components of the sample. The Schotten–Baumann reaction (SB reaction) is a commonly used procedure for the derivatization of primary, secondary, and tertiary amines in GC/NPD, GC/FPD, and GC/MS. Various alkyl chloroformates as derivatizing agents have been tested for these purposes; their utilization in aqueous solutions and two-phase solvent solutions have been reported and reviewed [23,24,25]. Aniline and its substituents are among the simplest weak bases that are highly susceptible to electrophilic and nucleophilic substitutions as the basis of SB reactions. For chloro- and bromo-substituted derivatives, a characteristic isotope pattern can be successfully used to identify fragments. This is why these substances have been chosen as potential derivatizing agents to analyze non-ionizable rhenium complexes. The resulting derivative contains a nitrogen atom as an easily ionizable group and improves the ionization efficiency of the analytes by ESI. 

Recently, we showed the suitability of MS for structure characterization of Re^V, VI, VII^ complexes with aromatic bidentate ligands [26]. Although MS provided useful structural information in a series of observed reaction and degradation products, we have occasionally met hardly ionizable structural types [27]. Namely, Re^VII^ oxochloro catechol complex resisted against ionization under ESI, APCI, and APPI conditions. We found an immediate solution in a reaction with *p*-bromoaniline and assumed this reaction worthwhile for further investigation and optimization. This contribution aims to systematically investigate the possibility of chemical conversion of bis(1,2-dihydroxybenzen)-chloro-oxorhenium complex to ESI ionizable products via reactions with aniline and its *p*-substituted derivatives (*p*-chloroaniline, *p*-bromoaniline, *p*-isopropylaniline, *p*-toluidine. The time-course of the derivatization reaction was followed both by ESI/MS and UV-Vis kinetic measurements. Collision-induced dissociation experiments revealed typical fragmentation pathways of formed molecular anions. 

## 2. Results and Discussion

### 2.1. Chemistry 

The hardly ionizable Chlorocomplexes [Re^VII^(O)Cl(Cat)_2_] were prepared using a procedure that was adapted from [26]. One equivalent (4.4 mg) of tetrabutylammonium tetrachlorooxorhenate(V) [(n-Bu_4_-N)(ReOCl_4_)] was dissolved in 1.5 mL of acetonitrile. Two equivalents of ligand 1,2-dihydroxybenzene and two equivalents of triethylamine (10% (*v*/*v*) solution in acetonitrile) were added and the reaction mixture was stirred for 3 days at laboratory temperature. Since catechol ligand lacks the free dissociable group, the yielding compound remains uncharged and its structural characterization by MS is impossible. Its presence in the reaction mixture was presumed entirely from a similarity between absorption spectra describing the formation of deprotonated pyrogallol analog. However, its ESI-MS structure identification is possible after the reaction with aniline (Figure 1), yielding an ESI ionizable reaction product [27]. 

Derivatives were prepared by mixing 10 μL of the reaction mixture described above with 50 μL of *p*- substituted aniline (5% (*v*/*v*) solution in acetonitrile). The reaction scheme of derivatization is shown in Figure 1. All prepared complexes are described in Table 1. For full chemical names see Appendix A.

### 2.2. Molecular Modeling

Density functional theory could be a possible MS/MS prediction tool useful in structure elucidating. Molecular modeling proposed the significant change of molecular conformation of studied aniline derivatives before and after ionization. We illustrate the structures of neutral and deprotonated [Re(O)(Cat)_2_PBrA] in Figure 2.

Evident is the shortening of the Re-N bond in the course of deprotonation. As it results from more detailed modeling, the shortening of the Re-N bond is accompanied by the prolongation of the Re-O(20) bond in a ligand, the increase in the Re-N-C(22) angle, and the decrease in the dihedral angle C(14)-O(22)-Re-O(19). For details, see the corresponding computation data in the Appendix A. The extent of such structural changes depends on the aromatic substituent linked via derivatization. The decisive element here seems to be the basicity of the aniline derivative entering the SB reaction. The bar graph showing the correlation between Re-N bond shortening and the basicity of the used aniline derivative is shown in Figure 3. The pKa values of used aniline derivatives were obtained from [28].

### 2.3. High-Resolution Mass Spectrometry Characterization

All molecular ions of prepared derivatives exhibit characteristic isotopic distributions. We compared the similarity between the calculated and experimental molecular ion isotopic patterns using the similarity index (SI) [29]. The [(1-SI) 100] values given in Table 2 indicate an apparent coincidence between the calculated and experimental spectra and the correct assignment of the elemental composition to the *m*/*z* values of the observed ions.

Excellent agreement between the isotope pattern calculated and that obtained by MS of the [Re(O)(Cat)_2_PBrA]- complex is documented in Figure 4. An analogous satisfactory agreement of exact mass and isotopic distribution for all other derivatives was observed.

### 2.4. Reaction Time-Course

A significant color change accompanies the reaction of the [Re^VII^(O)Cl(Cat)_2_] complex with the aniline derivative. Therefore, UV-Vis absorption spectrophotometry can be applied for the evaluation of its reaction rate. The major absorption band at λ_max_ = 630 nm and the minor one at λ_max_ = 390 nm characterize the intensive blue-green coloration of the [Re^VII^(O)Cl(Cat)_2_] complex. The product of the reaction with the aniline derivative is pale yellow, showing an absorption maximum at 355 nm. The rate of the derivatization reaction with *p*-bromoaniline was followed by the UV/Vis absorption measurement. The corresponding spectra recorded at defined time intervals are shown in Figure 5. A decrease of more than 90% over the 15 min interval was observed for the absorption band at λ_max_ = 630 nm. The proportional decrease in the height of the related absorption band at λ_max_ = 390 nm was observed. After 50 min, both absorption maxima disappear in favor of a minor absorption band at λ_max_ = 355 nm (see inset).

An alternative insight into the mechanism of derivatization reaction has been provided by complementary ESI/MS kinetic measurement displayed in Figure 6. The intensity of ion *m*/*z* 588 (red line), as the desired derivatization product, achieves a maximum at approximately 15 min. This is consistent with the observed rate of the decrease in the [Re^VII^(O)Cl(Cat)_2_] absorption band (λ_max_ = 630 nm). The appearance of ion *m*/*z* 454 (blue line) revealed another function of aniline derivative acting as a weakly basic accelerator of the Re^V^complex oxidation to a higher Re^VI^ form. Therefore, we can observe a decrease in the intensity of peak *m*/*z* 419 (black line) due to the slow transformation of complex [Re^V^ (O)(Cat)_2_]^−^ present in the reaction mixture. As the Re^VI^ complex species are prone to further oxidation, the intensity of ion *m*/*z* 454, reaching a maximum within 15 min, decreases at a moderate rate.

The time-course of ionic intensities helps to reveal the actual structure of ion *m*/*z* 454. Although the high-resolution MS data are available, there is still uncertainty about the exact structure of ion *m*/*z* 454, where both [Re^VI^ (O)Cl(Cat)_2_]^−^ and the adduct with chlorine [Re^VII^ (O)(Cat)_2_]Cl^−^ fit in the same mass. We believe that the shape of the time dependence of the ionic intensities presented in Figure 6 precludes the presence of a chlorine adduct.

### 2.5. Collision Induced Dissociation (CID)

The tandem mass spectrometry at 35eV collision energy in negative ionization mode was used to determine the structure of the prepared derivatives. The obtained MS/MS spectrum in Table 3 is very simple. The accurate mass measurement indicates that the main peak *m*/*z* 480 is formed by the loss of catechol moiety from the precursor marked by the blue diamond. It is evident that the isotopic profile retains the distribution confirming the presence of rhenium and bromine isotopes. It is even possible to observe the loss of the aromatic ring from the other catechol moiety to form ion M-2L *m*/*z* 404. Fragmentation behavior suggested the presence of a remarkably strong Re-N bond. On the other hand, the ion *m*/*z* 327 is formed by the loss of substituted aniline and the whole process ends in ReO_4_^−^ and ReO_3_^−^ ions resp. Since the same behavior was observed for almost all prepared complexes, the fragmentation pattern was demonstrated in this example only. The proposed fragmentation pathway is presented in Figure 7. High mass-accuracy measurements according to Table 3 and fragmentation patterns allowed us to identify the structure of all prepared derivatives.

According to the collision-induced dissociation results in Figure 8, it can be seen that the intensity of the product ion formed by the loss of one catechol ligand M-L (*m*/*z* 480) reaches a maximum at the same collision energy as the ion formed by simultaneous fragmentation of substituted aniline (*m*/*z* 327), but the intensity of this ion is only around 10% relative to the major peak. This is consistent with the observed shortening of the Re-N bond.

The green curve on the CID diagram describing the formation of the ion *m*/*z* 251 does not exhibit a significant maximum, and it is evident that the formation of that ion corresponds to different processes. The spotting of this ion already at zero collision energy can be attributed to the decomposition of the complex; for example, by air humidity. Another mechanism of *m*/*z* 251 ion formation is the loss of the aromatic ring from *m*/*z* 327. Dissociation of the Re-N bond and cleavage of aniline from the *m*/*z* 404 ion occurs only at high CE.

Calculated significant deformation of the molecule, which should be linked both with the shortening of the Re-N bond and the prolongation of a Re-O(20) bond (approximately 0.2 Å) in a single ligand, has an experimental consequence in CID experiments. Such a bond is going to be the most easily fragmented, yielding ion *m*/*z* 480. The intensity of the simultaneously arising ion *m*/*z* 327 is then significantly lower due to the lower energetical convenience of such fragmentation pathway. As expected, the ion on the favorable fragmentation pathway has the highest negative energy. The difference in energies is about 177 kcal mol^−1^ in favor of ion *m*/*z* 480. The energy differences were converted from Hartrees into kcal mol^−1^ using a conversion factor of 627.5.

The fragmentation of all prepared complexes was analogous. The data are available in the Appendix A. We observed the only exception for [Re(O)(Cat)_2_PIPA]^−^. Here, the different behavior is not related to the change of bond length but the product stability of arising ions. As is evident from the corresponding CID diagram (Figure 8B), a significant shift to higher collision energies has been observed for the ion ReO_3_^−^. The collision energy where the ion *m*/*z* 235 reaches a maximum is almost 40 eV or 3.2 eV regarding E_CM_ higher. Unlike the other complexes, the ReO_3_^−^ ion (*m*/*z* 235) in a [Re(O)(Cat)_2_PIPA]^−^ fragmentation pathway is not formed directly from ion M-2L but through the ion *m*/*z* 325 as an intermediate. Ion *m*/*z* 325 is formed by a loss of the methane molecule (Figure 9), where aryl-vinyl stabilization due to π electrons conjugation takes part. Such types of stabilization are unique for [Re(O)(Cat)_2_PIPA]^−^ and not possible for other prepared complexes. We calculated that the difference in energies of the fragments with and without aryl-vinyl stabilization is around 48 kcal mol^−1^. The elemental composition of ion *m*/*z* 325 was verified using HRMS. We have obtained the exact mass of 351.9993 Da by measuring, while the calculated value for C_8_H_7_NO_3_Re^−^ is 351.9989 Da. It means the error is −1.0 ppm.

## 3. Materials and Methods

### 3.1. Materials and Reagents

Tetrabutylammonium tetrachlorooxorhenate(V), 4-Chloroaniline, 4-Bromoaniline, 4-Methylaniline, 4-isopropylaniline, aniline, 4-Methylcatechol, and 1,2-dihydroxybenzene were purchased from Sigma-Aldrich. Acetonitrile (HPLC grade) and triethylamine were purchased from Fisher Scientific. Nitrogen used as nebulizing and drying gas was generated by MS-NGM 11 (Bruker Daltonics, Bremen, Germany) nitrogen generator.

### 3.2. Instrumentation and Software

ESI/MS experiments were conducted on a Bruker QqTOF compact instrument operated using Compass otofControl 4.0 (Bruker Daltonics, Bremen, Germany) software. Compass DataAnalysis 4.4 (Build 200.55.2969) (Bruker Daltonics, Bremen, Germany) software was used for data processing. Molecular structures and fragmentation schemes were drawn using ChemDraw (PerkinElmer Informatics, Waltham, MA, USA). Isotope patterns and exact masses of ions were calculated using IsotopePattern 3.0 (Build 201.9.27) (Bruker Daltonics, Bremen, Germany) utility. Analytical scales Kern ALJ 220-4 (Kern & Sohn, Balingen, Germany) were used to weigh solids. Stirring procedures were performed using a Stuard SA8 (Cole Parmer, UK) stirrer.

ESI/MS data were collected in negative ion mode at scan range from *m*/*z* 50 to *m*/*z* 1000. The temperature of the drying gas was set to 220 °C at 3.0 L min^−1^ flow rate. Cone voltage was 2800 V. Samples were injected into the nebulizer by a syringe pump (Cole Parmer, USA) at a flow rate 3 µL min^−1^.

Time-based ESI/MS measurement was performed by mixing the reactants in concentrated form and diluting the reaction mixture right before the ESI ion source using the second syringe pump with acetonitrile.

The isolation width of parent ions in CID experiments was set to 5 Da, the pressure of collision gas (nitrogen) in the collision cell was 2.5 × 10^−3^ mbar. Measurements were conducted in the range from 10 eV to 200 eV collision energy (ELAB) with a step of 1 eV. Mass spectrometer was calibrated using clusters of ammonium formate. OriginPro 9.0 was used for fitting CID dependences.

Agilent 8453 spectrophotometer (Agilent, Santa Clara, CA, USA) was used for UV/Vis kinetic measurements. Spectra were recorded at 0.1 nm resolution from 330 to 750 nm and processed with UV-Visible Chemstation (Agilent, Santa Clara, CA, USA).

### 3.3. DFT Calculation

Optimizations of molecular geometries and other theoretical calculations were performed at the density functional level of theory (DFT B3LYP) using Gaussian 16 [30] and LanL2DZ basis set. Both non-ionized and ionized forms of studied complexes were optimized, and frequency calculations were performed with optimized structures at the same level of theory.

## 4. Conclusions

The Schotten–Baumann (SB) reaction has been successfully adapted for the derivatization of MS hardly ionizable Re(VII) chlorocomplexes. We systematically studied the reaction of the Re(VII) bis(catechol) chlorocomplex with the set of halogen and alkyl anilines as derivatization agents. The SB reaction products are easily ionizable under common ESI conditions providing structurally characteristic molecular and fragment anions. Based on DFT computation, the effect of Re-N bond shortening in the course of complex deprotonation was simulated and also correlated with the basicity of aniline derivative used as a derivatization agent. Our conclusions follow the known relation between the basicity of the reaction environment and the yield of the SB reaction. However, an attempt to increase the yield of the derivatization reaction by adding triethylamine (TEA) to the reaction mixture was unsuccessful. Such a conclusion probably refers to the fast reaction providing the dioxorhenium complex as a competition to the SB reaction itself.

The shortening of the Re-N bond throughout neutral molecule deprotonation and concurrent prolongation of Re-O(20) makes the cleavage of one or both ligands the most probable initial fragmentation pathway of studied complexes, leading to the formation of abundant M-L and M-2L anions.

Although the fragmentation of all studied complexes was analogous, we observed a notable difference concerning the formation of ReO_3_^−^ (*m*/*z* 235) ion in a fragmentation scheme of [Re(O)(Cat)_2_PIPA]^−^. Unlike the other complexes, this ion is not formed directly from an M-2L fragment but through an aryl-vinyl stabilized unique *m*/*z* 325, unseen in other studied complexes’ fragmentation schemes.

## Figures and Tables

**Figure 1 molecules-26-03427-f001:**
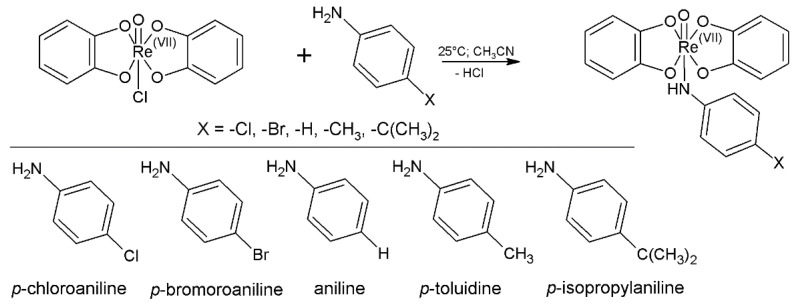
Derivatization reaction of uncharged rhenium(VII) chlorocomplexes with *p*-substituted aniline.

**Figure 2 molecules-26-03427-f002:**
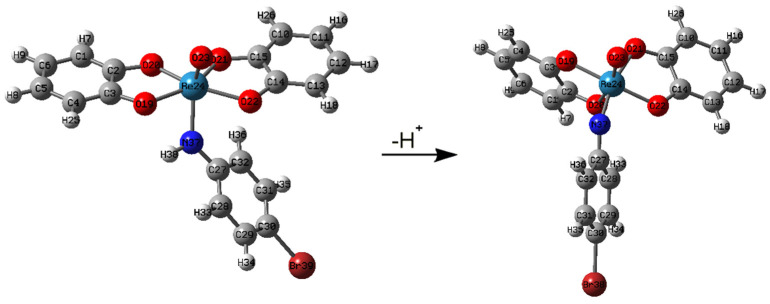
The DFT calculated structures of neutral and deprotonated [Re(O)(Cat)_2_PBrA].

**Figure 3 molecules-26-03427-f003:**
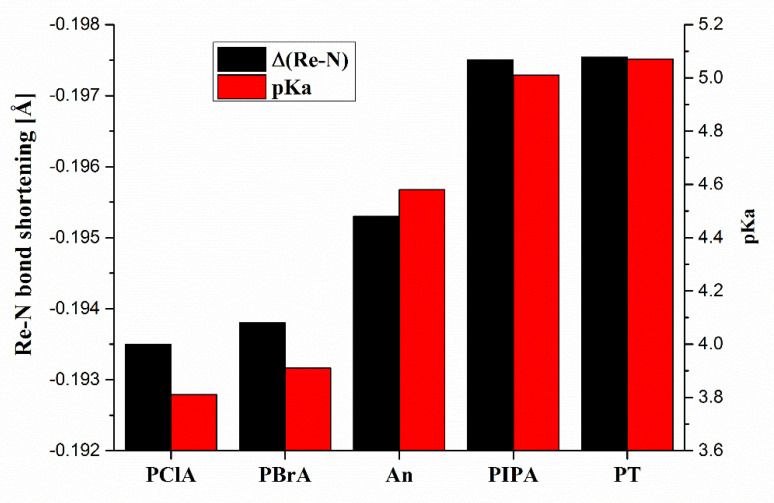
Correlation between Re-N bond shortening Δ(Re-N) and pKa of *p*-substituted anilines; PClA-*p*-chloroaniline, PBrA-*p*-bromoaniline, An-aniline, PIPA-*p*-isopropylaniline, PT-*p*-toluidine.

**Figure 4 molecules-26-03427-f004:**
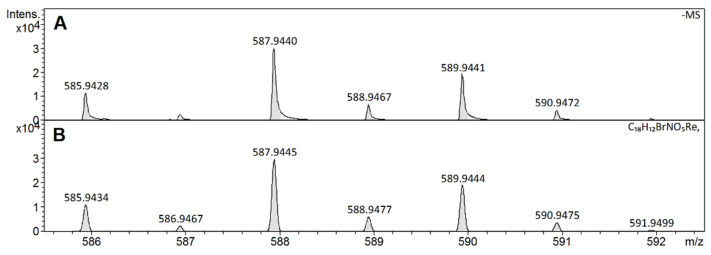
Experimental (**A**) and calculated (**B**) isotope pattern of [Re(O)(Cat)_2_PBrA].

**Figure 5 molecules-26-03427-f005:**
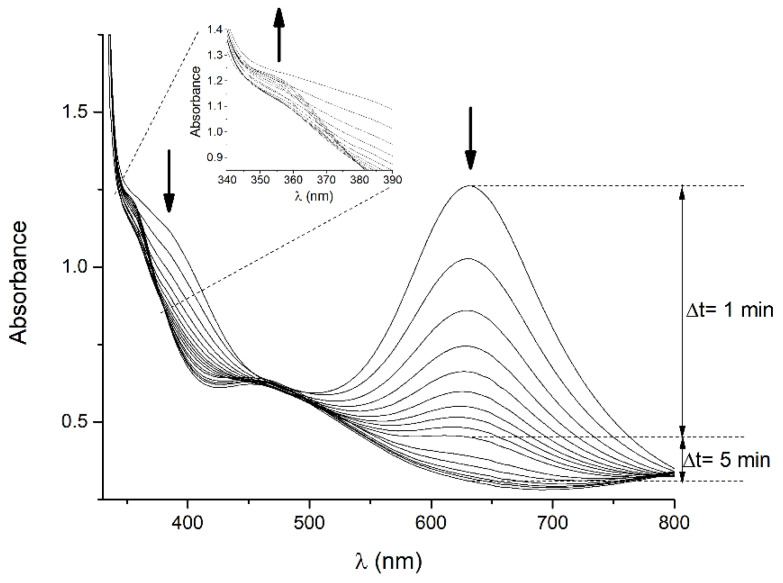
UV/Vis kinetics of derivatization reaction of uncharged [Re^VII^(O)Cl(Cat)_2_] chlorocomplex with *p*-bromoaniline.

**Figure 6 molecules-26-03427-f006:**
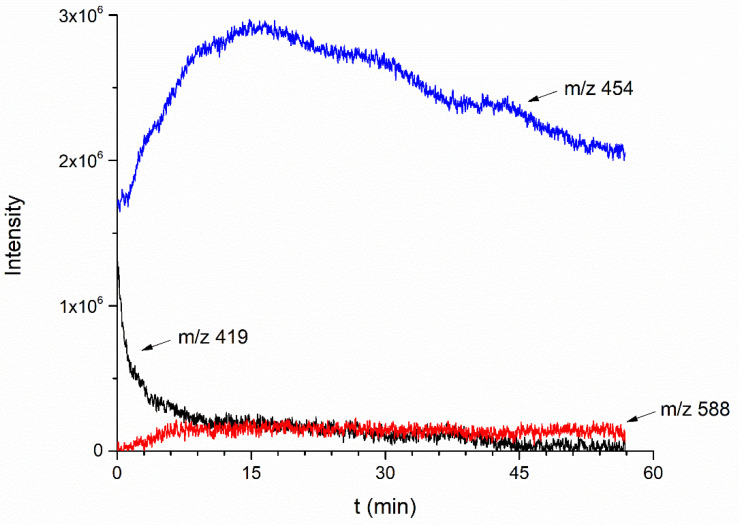
ESI-MS kinetics of derivatization reaction of uncharged [Re^VII^(O)Cl(Cat)_2_] chlorocomplex with *p*-bromoaniline.

**Figure 7 molecules-26-03427-f007:**
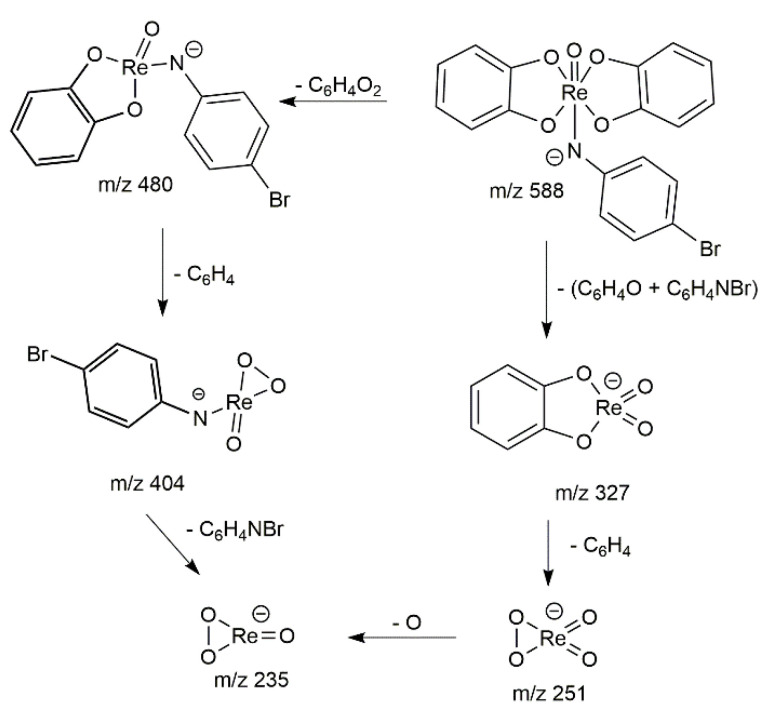
Fragmentation scheme of [Re(O)(Cat)_2_PBrA].

**Figure 8 molecules-26-03427-f008:**
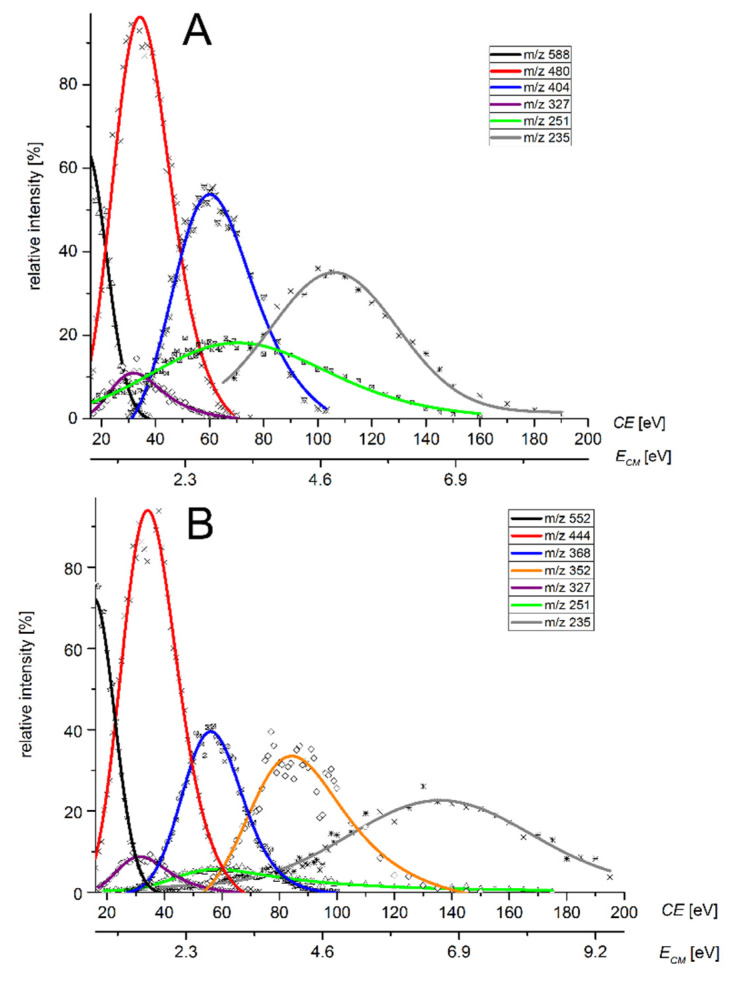
CID diagram of the dependence of the relative intensity of fragmented ions on the collision energy: (**A**) [Re(O)(Cat)_2_PBrA]^−^ (**B**) [Re(O)(Cat)_2_PIPA]^−^.

**Figure 9 molecules-26-03427-f009:**
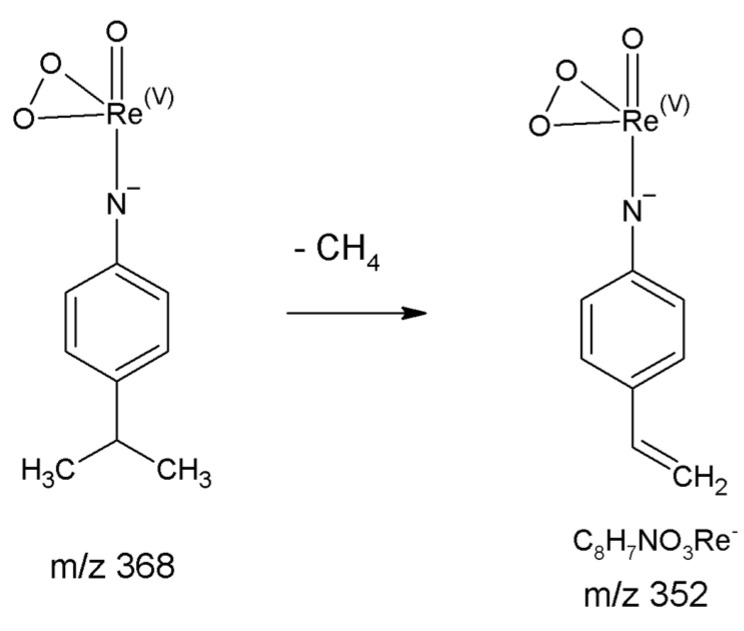
Proposed fragmentation mechanism of ion M-2L yielding from [Re(O)(Cat)_2_PIPA] complex.

**Table 1 molecules-26-03427-t001:** Labels and formulas of prepared rhenium complexes.

Entry	X	Complex	Formula
1	Cl	[Re^VII^(O)(Cat)_2_PClA]^− a^	C_18_H_12_ClNO_5_Re
2	Br	[Re^VII^(O)(Cat)_2_PBrA]^− a^	C_18_H_12_BrNO_5_Re
3	H	[Re^VII^(O)(Cat)_2_An]^− a^	C_18_H_13_NO_5_Re
4	CH_3_	[Re^VII^(O)(Cat)_2_PT]^− a^	C_19_H_15_NO_5_Re
5	C(CH_3_)_2_	[Re^VII^(O)(Cat)_2_PIPA]^− a^	C_21_H_19_NO_5_Re

^a^ Deprotonated ion.

**Table 2 molecules-26-03427-t002:** Theoretical and experimental exact masses of molecular anions of studied derivatives. The error values express the difference between theoretical and experimental masses; the similarity index expresses the resemblance between theoretical and experimental isotope patterns of molecular anions.

Entry	Molecular Formula	Theoretical *m*/*z*	Measured *m*/*z*	Error (mDa)	Error (ppm)	(1-SI) 100 (%)
[Re(O)(Cat)_2_PClA]^−^	C_18_H_12_ClNO_5_Re	543.9958	543.9969	−1.1	−2.0	98.6
[Re(O)(Cat)_2_PBrA]^−^	C_18_H_12_BrNO_5_Re	587.9445	587.9440	0.4	0.8	94.2
[Re(O)(Cat)_2_An]^−^	C_18_H_13_NO_5_Re	510.0357	510.0360	−0.3	−0.6	98.1
[Re(O)(Cat)_2_PT]^−^	C_19_H_15_NO_5_Re	524.0514	524.0507	0.7	1.2	89.2
[Re(O)(Cat)_2_PIPA]^−^	C_21_H_19_NO_5_Re	552.0827	552.0828	−0.2	−0.3	98.1

**Table 3 molecules-26-03427-t003:** Theoretical and experimental masses of [Re(O)(Cat)_2_PBrA] CID fragment ions measured at collision energy 35 eV. The error values express the difference between theoretical and experimental masses.

Nom. *m*/*z*	Ion Formula	Theoretical *m*/*z*	Measured *m*/*z*	Error (mDa)	Error (ppm)	Rel. Abundance (%)
588	C_18_H_12_BrNO_5_Re^−^	587.9462	587.9459	0.5	0.3	0.3
480	C_12_H_8_BrNO_3_Re^−^	479.9233	479.9271	−7.9	−3.8	−3.8
404	C_6_H_4_BrNO_3_Re^−^	403.892	403.8931	−2.8	−1.1	−1.1
327	C_6_H_4_O_4_Re^−^	326.9673	326.9681	−2.5	−0.8	−0.8
251	ReO_4_^−^	250.936	250.9354	2.3	0.6	0.6
235	ReO_3_^−^	234.9411	234.9383	11.7	2.8	2.8

## Data Availability

Data is contained within the article or Appendix A.

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
