# Peer review of "Chemical Conversion of Hardly Ionizable Rhenium Aryl Chlorocomplexes with p-Substituted Anilines"

_molecules, 2021, doi:10.3390/molecules26113427_

Round 1
Reviewer 1 Report
The authors describe a method for "derivatization of hardly ionizable rhenium aryl chlorocomplexes with aniline and its derivatives". To be honest, I don't quite understand the main focus of this work, in my opinion that manuscript should be completely rewritten. The introduction goes into points that are not relevant to the topic e.g. lines 71-72 and 91-92, moreover what do the authors mean with lines 107-109: "with the electron π of the benzene nucleus" and "the electronic charge is transferred to the d-orbital of the bromine atom present in the para-position"? The 3d orbitals in Br are filled (3d10) and cannot accept any electrons and 4d orbitals are irrelevant. The results described in section "DFT molecular modeling" are not relevant for the topic of that investigations and could be removed consequently. Moreover the acronyms given at the abscissa are not explained. Section 2.3 "Reaction Kinetic": The quantitative description of the reaction kinetics is missing.
I may recommend to thoroughly revise that manuscript prior to resubmission.
Reviewer 2 Report
In this study, Shotten-Baumann (SB) reaction was adapted for the derivatization of MS hardly ionizable Re(VII) chlorocomplexes.
In my opinion, the study was systematic and organized. The experiments were carefully executed and the reported data appeared to be overall solid. Data analyses were logical and well presented. Results obtained were interesting and were consistent with the conclusions.
In summary, the paper was put together well, contained a plethora of interesting new data hence it will be of interest to other researchers working in this area as well as to the readership of this journal.
Author Response
We found and fixed some spelling mistakes.
Reviewer 3 Report
The manuscript “Derivatization of Hardly Ionizable Rhenium Aryl Chlorocomplexes with Aniline and its Derivatives” by Sticha et al. is an interesting piece of work describing the use of the Schotten-Baumann reaction for the functionalization of Re(VII) complexes, in order to increase their ionizability. Mass spectrometry and DFT calculations were used to determine the structures of the synthesised compounds. The work presented here is solid, however, there are some points which need to be addressed before this manuscript can be published in Molecules.
1.) The title of the manuscript uses the words “derivatization” as well as “derivatives” which could be improved to make it more appealing. A possible suggestion is to replace the first “derivatization” with “functionalization”, but maybe the authors can come up with something more suitable.
2.) The abstract does not summarize the findings and outcomes of the manuscript. It seems rather generic. The first sentence of the abstract mentions radiopharmaceuticals while the entire manuscript is about mass spectrometry. This needs to be updated.
3.) The introduction part is quite long, but very interesting. Unfortunately, there are many abbreviations (e.g. GC-MS, ESI/MS, APCI, GC/NPD, GC/FPD, GC/MS, APPI), which are not explained. Especially the GC/NPD, GC/FPD, GC/MS abbreviations are rather specialised techniques and might not be well-established knowledge for all readers.
4.) The authors mention that they used the Re(VII) oxochloro catechol complex together with p-bromo aniline as an initial reaction to improve the ionization conditions of Re(VII) complexes (page 3, line 99ff). It would be helpful for the reader to move Figure 9 closer to this paragraph, in order to illustrate the outcome of the study.
5.) After reading the final part of the introduction, it is not clear, what the authors are going to show. The sentence “This contribution aims to investigate the possibility of chemical conversion of selected Re(VII) arylchloro complexes to ESI ionizable products via reactions with aniline derivatives.” leaves the reader with more questions then answers. (i) which selected Re(VII) arylchloro complexes are used? (ii) can the authors include a Scheme here? (iii) which aniline derivatives are used?
6.) The results of the manuscript are very interesting and show that it is possible to functionalize the Re(VII) catechol complexes. Unfortunately, it is very hard to follow the “Results and Discussion” part. It would be helpful for the reader to show an overview at the beginning (before 2.1), to highlight which compounds the authors are going to discuss.
7.) The DFT calculations part (2.1) only takes into consideration the para-bromo aniline compound (Figure 1), followed by Figure 2, where various aniline derivatives are compared without introducing them or explaining the different abbreviations. Especially the bar graph labels need improvement or explanation. What does “PClΛ, PBrΛ, Λn, PIPΛ, PT” stand for? Maybe it would be helpful to include small schematic drawings of the used aniline derivatives.
8.) Table 1 + 2. The entries need to be updated to replace the commas with full stops.
8.) It would be helpful to not abbreviate headlines e.g. HRMS
9.) In part 2.3, the authors mention a colour change (“change of coloration” should be replaced by “colour change”) of the Re(VII) complex with one aniline derivative (page 5, line 151). It is not clear, whether the authors have performed the UV-Vis absorption measurements with only p-bromo aniline (which the authors only can assume from the caption of Figure 5), or with all derivatives (where the authors still don’t know which ones were prepared by the authors).
10.) page 8, line 204 - 208: please use less abbreviations, it is really difficult for the reader to follow: replace CID with collision induced dissociation and CE with collision energy.
11.) The Figure caption of Figure 7 needs to be updated. It is not clear, which compounds were used to produce Figure 7 A or Figure 7 B.
12.) page 11, line 277-291. The authors left parts of the template in the submitted manuscript. This needs to be corrected.
13.) Synthetic part: The authors describe the synthetic method of the 7 produced compounds. There are no additional characterization methods mentioned which could help to clearly determine the formation of the proposed complexes: (i) Have the authors tried to crystallize the synthesised compounds? (ii) Is it possible to perform an elemental analysis of the compounds, to gain information about the composition? (iii) IR spectroscopy can show the formation of coordinated anilines.
14.) Supporting Information. The authors have prepared additional supplementary material, but there are no references in the main text to that material. The HR ESI is only recorded for complexes 3 - 7, but not for complexes 1 and 2. It would be helpful to sign the observed peaks to the proposed fragments.
In conclusion, the manuscript by Sticha et al. contains a lot of interesting information, but it is hard to follow the outline of the manuscript. The authors should therefore revise their manuscript, before it can be published.
Round 2
Reviewer 3 Report
Thank you for updating the manuscript. It looks nice now. I can recommend publication.